# Applications of Computer Vision in Monitoring the Unsafe Behavior of Construction Workers: Current Status and Challenges

**Wenyao Liu [1],\*, Qingfeng Meng [1], Zhen Li [1] and Xin Hu [2]**

[1] School of Management, Jiangsu University, 301 Xuefu Road, Zhenjiang 212013, China; mqf@ujs.edu.cn (Q.M.); janeli@ujs.edu.cn (Z.L.)

[2] School of Architecture and Built Environment, Deakin University, 1 Gheringhap Street, Geelong, VIC 3220, Australia; xin.hu@deakin.edu.au

\* Correspondence: 2221910009@stmail.ujs.edu.cn

**Abstract:** The unsafe behavior of construction workers is one of the main causes of safety accidents at construction sites. To reduce the incidence of construction accidents and improve the safety performance of construction projects, there is a need to identify risky factors by monitoring the behavior of construction workers. Computer vision (CV) technology, which is a powerful and automated tool used for extracting images and video information from construction sites, has been recognized and adopted as an effective construction site monitoring technology for the identification of risky factors resulting from the unsafe behavior of construction workers. In this article, we introduce the research background of this field and conduct a systematic statistical analysis of the relevant literature in this field through the bibliometric analysis method. Thereafter, we adopt a content-based analysis method to depict the historical explorations in the field. On this basis, the limitations and challenges in this field are identified, and future research directions are proposed. It is found that CV technology can effectively monitor the unsafe behaviors of construction workers. The research findings can enhance people's understanding of construction safety management.

**Keywords:** computer vision; construction workers; monitoring; unsafe behavior; literature review

## 1. Introduction

The construction industry is one of the most dangerous sectors in the world. Construction accidents cause deaths, injuries and other major direct and indirect losses of construction workers [1,2]. According to the statistics of the Ministry of Housing and Urban–Rural Development of the People's Republic of China (MOHURD), there were 773 production safety accidents related to housing and municipal engineering projects in China in 2019, which led to the deaths of 904 workers [3]. Occupational safety in the construction industry is a global problem, not unique to any country. According to the census data of the U.S. Bureau of Labor, there were 970 and 965 fatal construction accidents in the United States in 2016 and 2017, accounting for about 19% of all occupational deaths in that year [4]. In addition, the incidence of nonfatal occupational injuries and diseases in the construction industry is 30% higher than the industry average, especially for some fall injuries and musculoskeletal diseases [5]. Given the high incidence of fatal and nonfatal injuries in the construction industry, it is imperative to provide for effective safety management at construction sites [1].

Heinrich et al. [6] found that 88% of construction accidents are caused by the unsafe behavior of construction workers, while the rest of them result from the unsafe conditions of objects, which are also mostly caused by the unsafe behavior of workers. The "unsafe behavior" of construction workers refers to dangerous behavior that violates organizational discipline, operating procedures and methods in professional activities, and an

"unsafe state" refers to the material conditions that lead to accidents, including material and potential hazards in the working environment. These hazards are often caused by human operations; that is, the unsafe behavior of workers [7,8]. Consequently, the key to safety management at construction sites is to effectively manage on-site people and objects. Previous studies have shown that behavior-based security (BBS) is a widely used method in security research [9]. The use of BBS can help researchers to directly observe and identify people's unsafe behavior and eliminate these unsafe behaviors through feedback information [2,10]. Although BBS has achieved great success in the research field of construction safety management, this behavior measurement method, which mainly relies on human observation, has gradually shown many shortcomings. Han and Lee [11] summarized the three limitations of using BBS: (1) measurement is time-consuming [12]; (2) a large number of samples are needed to ensure the validity of conclusions [13]; (3) workers' active participation and manual observation are needed [14].

To solve these constraints and limitations, the use of computer vision (CV)-assisted technology is becoming popular. This technology provides an effective method to automatically capture and identify individuals' unsafe behavior at construction sites [10,11,15–17]. By using images or videos, CV technology can enhance project stakeholders' understanding of the information at construction sites, such as the location and movement status of workers and construction equipment. Compared with other sensor technologies (e.g., radio frequency identification technology (RFID), the Global Positioning System (GPS), ultra-wideband (UWB)), CV technology does not need to install sensors on each entity, which means savings in both time and cost. Additionally, given that CV technology is fast and accurate in detection, it has great potential for working as a safety and health monitoring tool at construction sites [18].

With the advancement of CV technology, an increasing number of researchers are using such technology to explore the topic of safety monitoring at construction sites. Seo et al. [18] made the first proposal for a general framework for computer-vision-based safety and health monitoring, which include object detection, object tracking and action recognition. This general framework provides a scene–location–action-based risk identification method. Target detection is a preliminary step of object tracking and action recognition. When the project entity appears in a scene, its spatial position can be tracked from continuous video frames according to the time progress using the object-tracking algorithm. The extracted position information can be used to identify unsafe conditions and behavior of entities. When there is a project entity with a cohesive structure (e.g., skeleton-based workers or component-based equipment), the action recognition technology will identify the posture of workers and equipment through static or continuous images to determine whether unsafe behavior exists or not. On the basis of this framework, Zhang et al. [19] divided the monitoring objects of CV into two aspects: (1) workers themselves and (2) the interactions between workers and the external environment. Fang et al. [10] reviewed the application of CV technology based on deep learning to monitor workers' unsafe behavior. Guo et al. [1] summarized the application of CV technology in the field of building health and safety monitoring, including monitoring workers and objects at construction sites (e.g., equipment, tools, resources) and construction activities (e.g., excavation, lifting, hoisting). Mostafa and Hegazy [20] pointed out that one of the main research directions of the image technology is for use in monitoring building safety, which mainly focuses on the three subtopics of the target detection technology used, the detected object and the resolution of the related security problems.

In this paper, we conduct a holistic literature review of the field relating to the use of CV technology in monitoring the unsafe behavior of workers at construction sites. On this basis, we identify the research gaps in the studied field and suggest corresponding future research directions to address these gaps. It is expected that the research will enhance construction stakeholders' understanding about the application of CV technology in monitoring the unsafe behavior of construction workers. In contrast to prior studies, such as [1], this research focuses more on the supervision of unsafe behavior of workers at construction

sites and reviews literature from the two perspectives of individual workers and worker–environment interactions. Additionally, unlike some historical studies (e.g., [10]) that only review the use of CV technology based on deep learning, this research examines the application of CV technology in a more comprehensive manner by using the traditional machine learning and deep learning methods.

This paper has six sections. The second section provides an overview of CV technology. In the third section, scientometric tools are adopted to summarize the historical explorations in this field. The fourth section, by using content analysis, provides a more detailed description about the studied field. On this basis, research discussions are provided and future research directions are proposed. In the final section, the research results and significance are summarized.

## 2. Background

### 2.1. Overview of Computer Vision

Computer vision (CV) is an interdisciplinary research field, and it mainly explores the methods to make a machine "see". Instead of using human eyes, CV technology uses cameras and computers to recognize, track and measure. It processes graphics into images that are more suitable for human eyes to observe or transmit to instruments for detection [10,21–23]. With the advancement of machine learning, computers have been trained to better understand what they "see". Machine learning focuses more on the methodology issues, while CV studies the application of technologies in real-world scenarios. Machine learning methods have been widely used in the CV field, such as the statistical machine learning represented by support vector machine (SVM) and the deep learning represented by artificial neural network (ANN) [24,25]. These two methods have played crucial roles in promoting the continuous development of CV technology in monitoring construction sites.

The original form of natural data processing process is cumbersome, which leads to the difficulties in achieving simplicity and automation. The traditional statistical machine learning method was widely used in the CV field [10]. Statistical machine learning relies on the preliminary understanding of data and the analysis of learning purposes. It uses engineering knowledge and expert experience to design feature descriptors, select appropriate mathematical models, formulate hyperparameters, input sample data and use appropriate algorithms for training and prediction. Its process is shown in Figure 1.

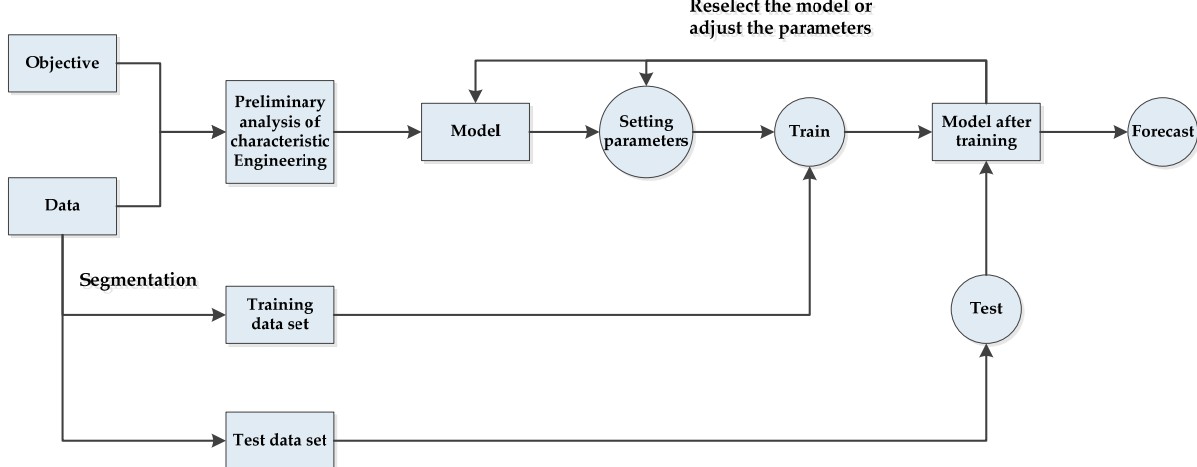

**Figure 1.** Basic flow chart of statistical machine learning.

To simplify the process of detection and recognition, an expression method based on deep learning (DL) has been developed. By learning from multiple data, this method can automatically extract complex features from end to end [25]. The structure of DL is comprised of layers (input layer, hidden layer, and output layer), neurons, activation function

"a" and weight {W, b}. Neurons play the role of feature detectors, and they are divided into low-level neurons and high-level neurons. The lower layers detect basic features and transfer them into higher layers before identifying more complex features [26]. The widely used deep learning methods in the construction safety field include convolutional neural networks (CNN) and recurrent neural networks (RNN) [26].

CNNs promote the development of image recognition technologies, and it is comprised of multiple layers of ANN [27]. Each layer of the network includes a two-dimensional plane, and each plane has multiple independent neurons. Besides the conventional input layer, output layer and activation layer, a CNN also has a convolutional layer and a pooling layer (as shown in layers 2 to 7 in Figure 2). The convolutional layer uses different two-dimensional filters and gradually slides to all positions of the two-dimensional image to achieve the inner product of the pixels of the image. The pooling layer is added after the convolutional layer. It reduces the output size of the convolutional layer by calculating the average and maximum values of the image at different pixels [27].

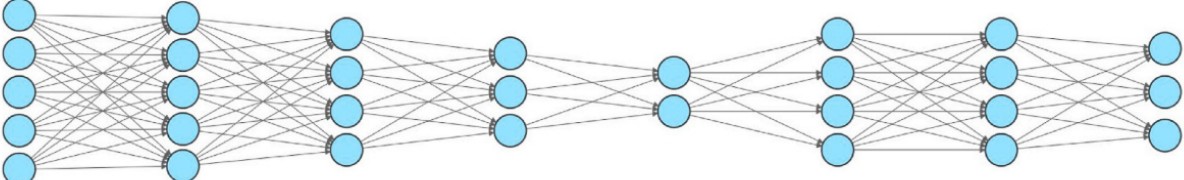

Input Layer ∈ $\mathbb{R}^5$ Hidden Layer ∈ $\mathbb{R}^5$ Hidden Layer ∈ $\mathbb{R}^4$ Hidden Layer ∈ $\mathbb{R}^3$ Hidden Layer ∈ $\mathbb{R}^2$ Hidden Layer ∈ $\mathbb{R}^4$ Hidden Layer ∈ $\mathbb{R}^4$ Output Layer ∈ $\mathbb{R}^3$

**Figure 2.** Convolutional neural networks (CNNs) architecture. Reproduced with permission from ref. [26]. Copyright 2020 Elsevier.

CNN can extract local features by adding a convolution operation to the neural network and obtain global features. On this basis, CNN uses a classifier to identify entities. CNN usually uses spatial characteristics (e.g., spatial locality) without considering temporal characteristics. However, a lot of real-world data are time-series-based (e.g., a piece of text), which means that these data must be organized in order and that the order cannot be randomly disrupted. Therefore, these data cannot be directly used and learned by CNN due to their temporal characteristics. As a result, RNNs that can process time series data are developed [28]. As RNNs add loops to the neural network, they have the advantage of limited short-term memory. Its structure is shown in Figure 3.

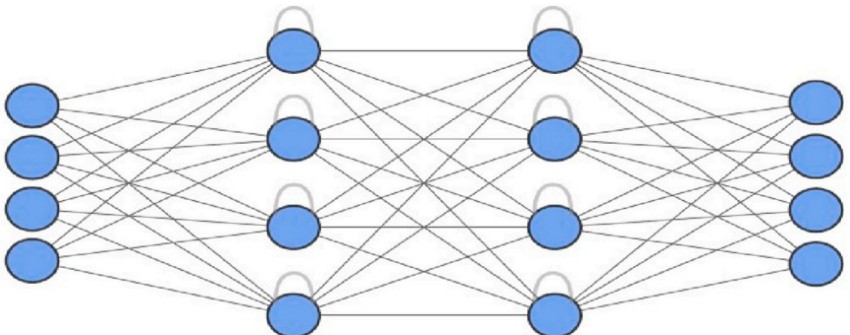

Input Layer ∈ $\mathbb{R}^4$ Hidden Layer ∈ $\mathbb{R}^4$ Hidden Layer ∈ $\mathbb{R}^4$ Output Layer ∈ $\mathbb{R}^4$

**Figure 3.** Recurrent neural networks (RNNs) architecture. Reproduced with permission from ref. [26]. Copyright 2020 Elsevier.

The traditional RNN model only has the function of short-term memory. However, many real-world scenarios, especially the scenarios at construction sites, are complex and changeable and require a network with the long-term memory function. Thus, the long

short-term memory (LSTM) model is developed [29]. At construction sites, researchers usually integrate CNN and LSTM to extract the spatial and temporal information of individual unsafe behavior (e.g., abnormal climbing and bending). The specific process is shown in Figure 4.

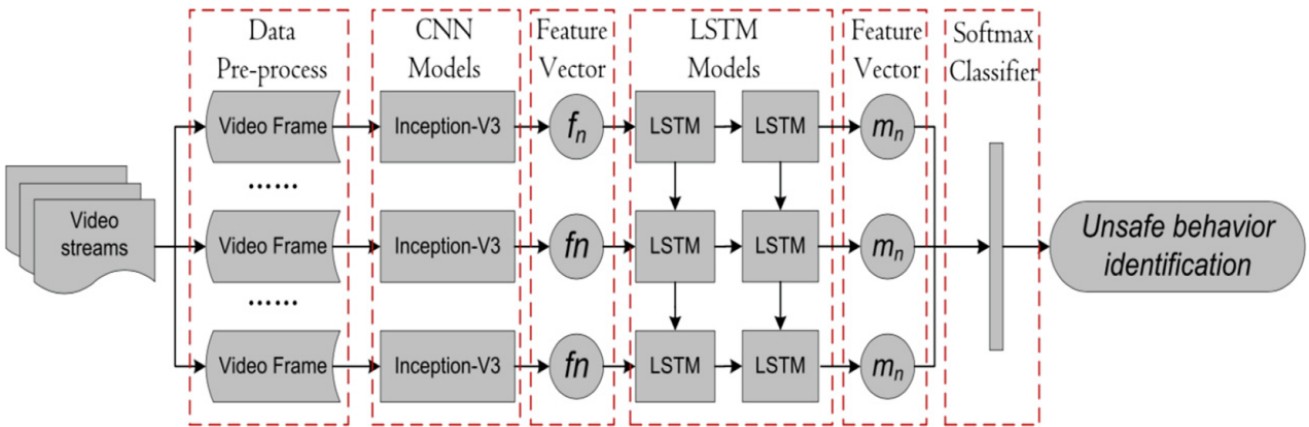

**Figure 4.** Example of using CNN-LSTM model to identify worker unsafe behavior. Reproduced with permission from ref. [30]. Copyright 2018 Elsevier.

### 2.2. Roles of Computer-Vision-Based Methods at Construction Sites

Currently, the research on CV technology in the construction industry mainly focuses on building structure monitoring and productivity analysis [26]. There is still a lack of research on identifying unsafe behavior by using such technology. The traditional identification and control of unsafe behavior mainly rely on manual methods. Nevertheless, the performance of manual methods is poor, especially given that a large number of images taken by the monitoring camera cannot be processed automatically and effectively. The development of CV technology provides support for the automatic identification of unsafe behavior. In particular, the CV technology does not need to attach equipment to workers. This not only helps to reduce costs and but also decrease the potential impacts on workers. At the same time, the CV technology can also process a large number of image data quickly. Therefore, the CV technology is suitable for construction sites. As mentioned above, the BBS method can recognize unsafe behavior through human observation and use feedback information to change the unsafe behavior so as to enhance safety performance. The feedback information relies on the perceptions and cognitive abilities of observers [31]. Observers understand the different construction scenes through their own perceptions, such as the recognition of human bodies and objects, and the visual processing of temporal and spatial relationships. The perceived information is compared with safety rules, policies and previous relevant experience, which helps to identify unsafe conditions and behavior. However, the CV technology is limited to extracting unsafe information and cannot be used to evaluate information to identify unsafe behavior and conditions. Therefore, the unsafe behavior monitoring method developed by using the CV technology should not only consider the extraction of construction information but also combine with existing policies and relevant experience [18]. This requires a more systematic framework to discuss how the CV technology is applied to the complex construction sites.

As there are diverse unsafe conditions and behavior at construction sites, and they have unique characteristics, different CV technologies need to be used. Seo et al. [18] classified CV-based methods into three categories, including scene-based methods, location-based methods, and action-based approaches. The corresponding CV technologies are object detection, object tracking and action recognition.

Firstly, the scene-based approach is used to understand and evaluate any potential risks in a static scene by examining the scene in a safe context. Scene understanding refers to the integration of the information of various components at construction sites [32]. Its

main purpose is to understand "what is in the scene (e.g., people, materials, machines, etc.)". Therefore, object detection technology is applied in this method. This technology searches the image through the known object model, and the object of interest can be detected based on the semantic information. Only when the project entity of interest is confirmed can follow-up in-depth research be carried out. In general, the scene-based approach is the first step, and it is also the cornerstone of the entire research [18]. For instance, it can be used to detect whether workers' safety protection equipment is in place and whether workers are working in an unsafe area [33,34]. Secondly, as the construction workers and equipment are dynamic and their positions change with time at construction sites, this requires the use of a location-based method to evaluate potential risks in different scenes. The location information of related entities can be obtained through tracking, which is of great importance to the identification of unsafe conditions and behavior, such as improper working positions (e.g., the proximity between equipment and workers) and incorrect equipment utilization (e.g., an excessive equipment speed) [18]. Finally, the action-based method focuses on the analysis of unsafe actions (e.g., bending, squatting, climbing, weight lifting) of construction workers. These actions are the main causes of workers' musculoskeletal diseases (MSDs) and ergonomic injuries [35]. The recognition of workers' actions helps to remind workers to improve their inappropriate work postures, which improves workers' health and safety.

In summary, CV based methods can be divided into three categories, including object detection, object tracking and action recognition. The use of these methods makes it possible to intelligently monitor unsafe behavior and conditions at construction sites.

Object detection can be used to identify unsafe behavior and conditions at construction sites. The most common method is to divide a captured large image window into small spatial areas for analysis. Features will be extracted from small areas, and the retrieved features can be classified [36]. Its speed and accuracy are constantly improving from manual extraction to automatic extraction and from SVM to CNN. The probability of discovering unsafe behavior is also greatly increased.

Object tracking can create the time track of detected objects when moving in the scene and identify its real-time position. There are two main kinds of research, including CV-based 2D tracking and 3D tracking [37]. 2D tracking mainly tracks a target by matching the feature points and shape contours in the video frame, while 3D tracking mainly uses 3D tracking sensors to establish 3D coordinates to obtain movement information (e.g., path, velocity, acceleration, direction, etc.) [18]. From the perspective of space, this method can comprehensively detect unsafe behavior of workers.

Action recognition is the process of labeling action labels on images. This method can extract human features from images, such as shape and time motion, which is conceptually similar to the feature extraction of target detection. But it is a more complicated process because some specific motion vectors are added (e.g., joint position, joint angle). This method has the advantage of better extracting small actions [35,38]. These three methods can monitor construction sites well, identify the unsafe behaviors of workers and make great contributions to the improvement of construction safety management.

## 3. Research Methods and Material Preparation

The aim of this study is to comprehensively reveal the research status of CV technology in the field of monitoring unsafe behavior of construction workers through a comprehensive literature review. This study adopted the comment method based on content analysis. This method is a recognized method of carrying out literature review through synthesizing findings of historical studies [19]. In this section, on the basis of a systematic bibliometric analysis, the academic relationships and research hotspots of CV in the field of building safety are mapped. In addition, the research theme is highlighted and determined, and the previous research framework and context are corroborated. In addition, the applicability and quality of the obtained literature are ensured through the selection of topics and

research fields and periodical screening. This provides a foundation for the content-based analysis in the next section.

### 3.1. Literature Search and Selection

A bibliometric search was conducted in the Web of Science (WOS) database. WOS has powerful analysis abilities, which can quickly locate high-impact papers and identify research directions concerned by global researchers, especially the Science Citation Index Expanded (SCIE) and Social Science Citation Index (SSCI) in the core collection of WOS. These two academic journal paper citation index databases contain the most comprehensive high-impacting academic journals in the world [39]. In addition, the conference proceeding Citation Index-Science (CPCI-S) in the core collection of WOS covers the annual meeting minutes of various industry authorities, which is also leading edge and guiding. Therefore, the SCIE, SSCI and CPCI-S databases in the core collection of WOS are used as reference sources. To ensure a comprehensive research result, the different keywords and Boolean operators "AND" and "OR" are adopted. Based on the "advanced search" function of WOS, the searching strategy used in this study is: "TS = ((construction worker *) AND ((safety) OR (risk) OR (health)) AND ((machine learning) OR (deep learning) OR (computer vision *) OR (vision-based)))". The search was limited to the time period 2000–2021. The search was conducted on March 1, 2021, and 134 papers were obtained, including journal papers and conference papers.

Criteria were also developed to select appropriate papers for this study. These criteria are: (1) a paper focusing on the health and safety monitoring of construction site workers; (2) a paper focusing on CV technology or technology integrated with CV; (3) a paper written in English. Finally, 122 papers were identified and used in this study.

### 3.2. Literature Analysis Based on Statistical and Bibliometric Tools

Firstly, the publication trend in years was analyzed (Figure 5). As shown in Figure 5, only a few papers were published in this field before 2016. Nevertheless, the increased research interest can be found after 2016. Especially, a larger number of papers were published in the field in 2018–2020, with the largest number of publications arriving at 35 in 2020. This trend indicates that the interest of exploring related topics in the studied field is increasing in recent years, which has been promoted by various factors, such as the continuous development of computer technologies (especially the application of deep learning) and the growing importance of "safe production" and "people-oriented".

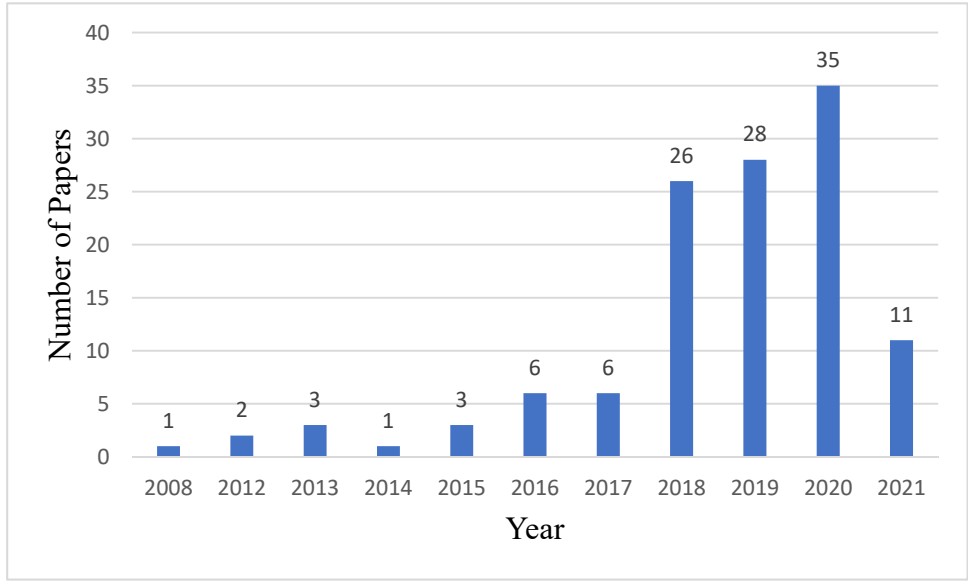

**Figure 5.** Number of papers published in different years.

This study also analyzed the publication sources of the used literatures (Figure 6). It can be seen from Figure 6 that most of the studies were retrieved from engineering management journals such as "Automation in Construction", "Advanced Engineering Informatics", "Journal of Construction Engineering and Management" and "Journal of Computing in Civil Engineering".

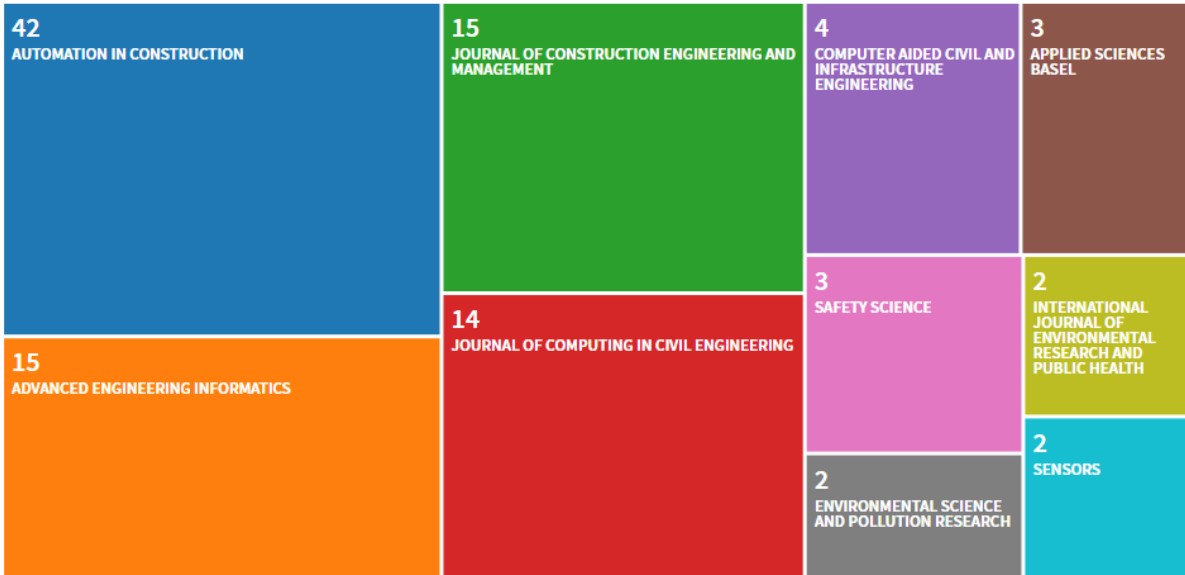

**Figure 6.** Source statistics of publications.

By using the visual bibliometric software of VOSviewer, the author cooperation network map in this field was developed (Figure 7). The node size indicates the number of papers, and the connection length indicates the degree of cooperation. In addition, a keyword hotspot map was also developed by using the VOSviewer (Figure 8). As shown in Figure 8, the research hotspots mainly include CV, deep learning, workers, safety, construction, equipment, recognition, tracking and identification. This result also confirms that the main research contents focus on "using the CV technology to detect, track, and identify workers and entities at construction site for safety prediction and prevention".

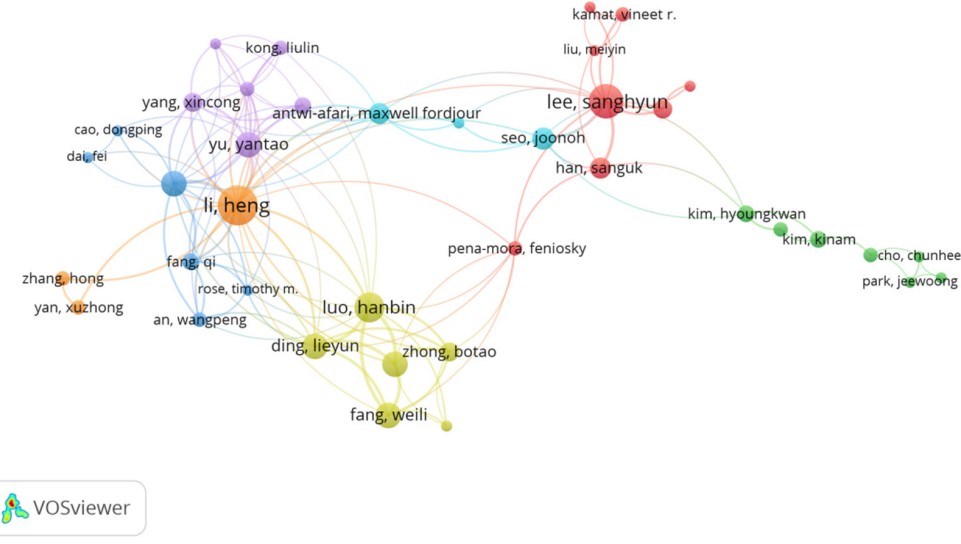

**Figure 7.** Author collaboration network.

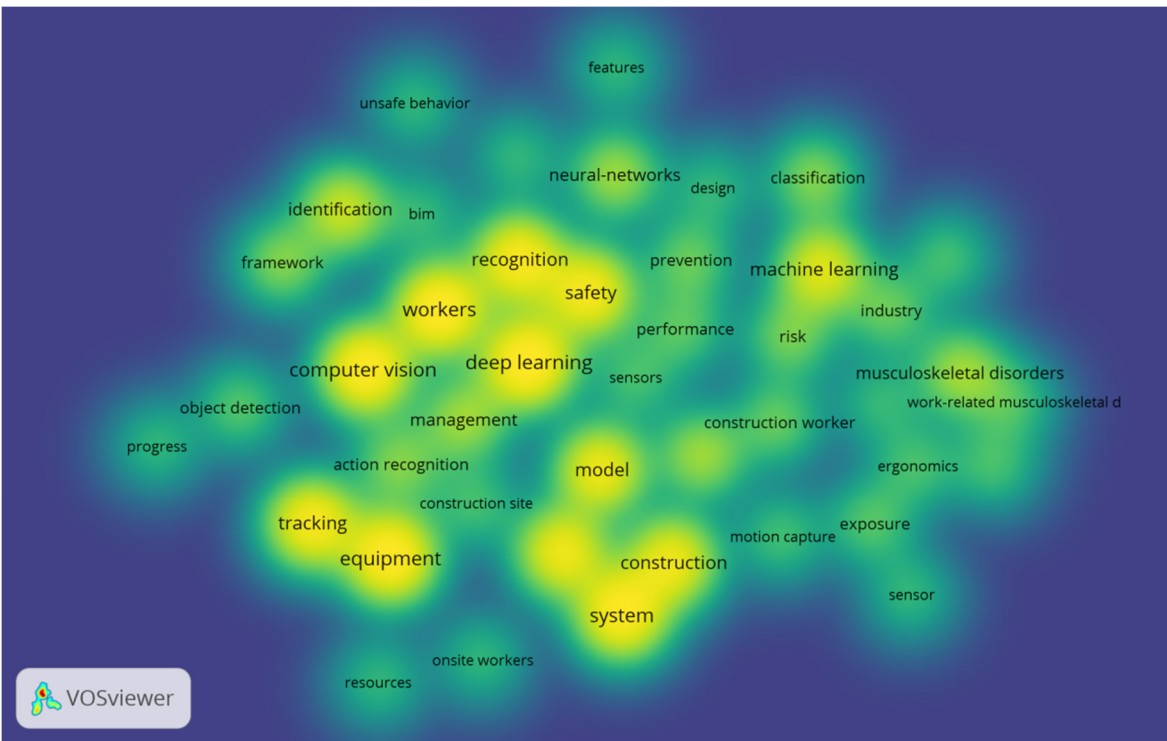

**Figure 8.** Keyword hotspot map.

## 4. Content-Based Literature Review

### 4.1. The Perspective of Workers Themselves

It is difficult to manage work-related factors, and these factors are one of the main causes of construction accidents and physical injuries. The application of CV technology to monitor workers mainly focuses on two aspects, including the detection of workers' use of personal protective equipment and the recognition of worker behavior and movements.

#### 4.1.1. Use of Personal Protective Equipment

When workers perform construction activities, they are surrounded by various risks, such as falling objects, construction equipment collisions and falls from heights caused by imbalance [19]. The appropriate use of personal protective equipment (PPE) has been confirmed as one of the effective methods to reduce construction incidents [40,41]. In the field of construction safety management, the current research mainly focuses on the detection of three types of equipment, including helmets, seat belts and safety vests. Researchers often use the image-based object detection technology to monitor the PPE use of construction workers.

Because deep learning has not been widely used, the PPE detection scheme based on image features mainly relies on the traditional statistical machine learning. Researchers generally use the gradient direction histogram (HOG) detector and the SVM classifier to detect and classify the PPE use of workers. The general process is divided into four steps, including detecting the human body, detecting the protective equipment (e.g., safety helmet), matching the detected human body with the equipment and evaluating the performance of the above three steps through measuring the detection accuracy and recall rate. Regarding the human testing, the HOG is the most popular and successful human body detector (Figure 9). The HOG uses "global" characteristics to describe a person instead of a collection of "local" characteristics. This means that a human body is represented by one feature vector instead of many feature vectors to represent smaller parts of the body. The HOG human detector uses a sliding detection window to move around the image and calculates HOG descriptors at each position of the detection window. Thereafter,

this descriptor is displayed to the trained classifier who classifies it as "human" or "non-human" [42]. The detection methods for PPE are diversified, and suitable methods can be selected for the detection of the salient features of protective equipment (e.g., shape, color). Common detection methods include HOG feature detection [16], color-based feature extraction, circular Huffman transform (CHT) [43] and HSV color detection [44]. By matching the detected human body with PPE, it can help to make the judgement whether a worker is wearing PPE correctly or not.

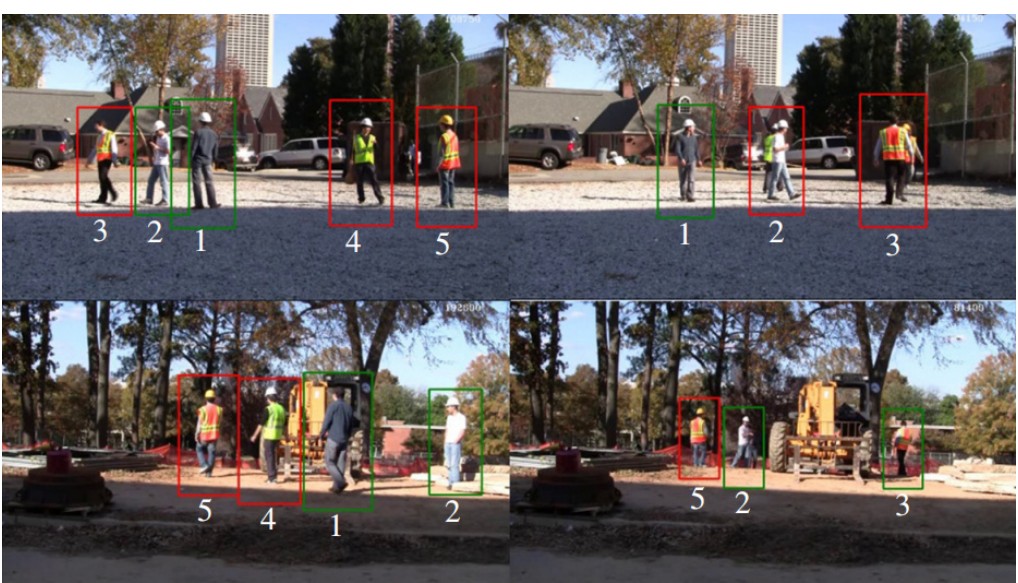

**Figure 9.** Example of the HOG-based human body detection in the foreground regions. Reproduced with permission from ref. [45]. Copyright 2012 Elsevier.

With the continuous development of computer technologies, the use of target detection technology that relies on deep learning is becoming more and more popular. It can be divided into two categories, including two-stage detection methods based on candidate regions and one-stage detection methods based on regression [36,46]. The two-stage methods include R-CNN, Fast-R-CNN, Faster-R-CNN and other detection methods. These methods need to generate candidate regions and classify and locate these candidate regions. A close examination of the historical studies found that the most used detection model is Faster-R-CNN. This model can ensure the accuracy of detection when facing constantly changing scenes and objects. Compared with traditional HOG + SVM, Faster-R-CNN has a short calculation time and can perform real-time detection. Fang et al. [15], Fu et al. [47], and Fang et al. [48] used the Faster-R-CNN model to optimize the convolution network structure and network training parameters in order to detect construction site staff and their protective equipment.

The one-stage methods mainly include single shot multibox detector (SSD) detection methods and YOLO series (YOLO, YOLO 9000, YOLO v3) detection methods. These methods can directly and simultaneously predict the category and location of targets by only using the CNN network, and they have shown good real-time performance. The network structure of the two-stage target detection algorithm that relies on the candidate area is complex. Although its detection accuracy is high, its detection speed is relatively slow. This shortage means that the two-stage target-detection algorithm cannot meet the real-time requirements of the construction industry. In contrast, the one-stage target-detection algorithm can complete the target-detection in time. For classification tasks, the entire network is only comprised of convolutional layers, and the input image passes through the network only once. This means that the detection speed is fast, which perfectly meets the real-time requirements of production practices [46]. Li et al. [49] proposed a CNN-based SSD-MobileNet algorithm to detect whether workers are wearing helmets

or not. Huang et al. [46] used the YOLO v3 algorithm to deal with the helmet-wearing problem of construction workers. Table 1 summarizes the research on PPE-use identification in workers.

**Table 1.** Research details of worker PPE-use detection.

| Reference | Object(s) | Algorithm Model | Methods | Contributions | Limitations |
|---|---|---|---|---|---|
| Park et al. [16] | hardhat | Statistical Machine Learning | (1) Human body detection (background subtraction + HOG feature) (2) Safety helmet detection (HOG feature) (3) Match between the detected human body and the helmet (4) Evaluate the detection performance through accuracy and recall rate | (1) Facilitate the safety monitoring work of the safety inspectors at the construction site (2) With an overall accuracy of 94.3% and a recall rate of 89.4% | (1) The detection template can only detect standing workers (2) The problem of occlusion |
| Rubaiyat et al. [43] | hardhat | Statistical Machine Learning | (1) Image segmentation (Gaussian mixture model GMM) (2) Human body detection (HOG) (3) Use color-based feature extraction and circular Hough transform (CHT) features for helmet detection (4) Classification (SVM) | (1) Safety helmets composed of specific colors such as yellow, blue, red and white can be detected (2) It can distinguish between ordinary hats and safety helmets | (1) The overall detection accuracy needs to be further improved, and deep learning techniques need to be used |
| Seong et al. [44] | vest | Statistical Machine Learning | (1) Color space (HSV) + classifier | (1) Use the color of the safety vest as a key feature for detecting, locating, tracking and monitoring workers | (1) Since only color detection is used, detection errors may occur |
| Fu et al. [47]; Fang et al. [48] | hardhat | Deep Learning | (1) Use Faster-R-CNN to automatically detect image features | (1) Real-time detection with high precision and high recall rate can be achieved in different scenarios, which can reach 95.7% and 94.9% respectively (2) It can effectively detect the staff in the far-field surveillance video | (1) When faced with problems of occlusion and weak light, the detection accuracy is very low |
| Fang et al. [15] | harness | Deep Learning | (1) Faster-R-CNN for detecting the presence of workers (2) Deep CNN used to identify the harness | (1) The detection accuracy is as high as 99% (2) Overcoming the difficulty of using the detection harness | (1) Affected by light and object occlusion |

**Table 1.** *Cont.*

| Reference | Object(s) | Algorithm Model | Methods | | Contributions | | Limitations | |
|---|---|---|---|---|---|---|---|---|
| Li et al. [49] | hardhat | Deep Learning | (1) | SSD-MobileNet algorithm based on CNN | (1) | The real-time performance and speed of detection have been greatly improved | (1) | When the image is not very clear, the helmet is too small or the background is too complicated, the detection performance is poor |
| | | | | | (2) | It does not require manual feature selection, has better image feature extraction capabilities, and has higher accuracy and recall | (2) | Affected by object occlusion |
| Huang et al. [46] | hardhat | Deep Learning | (1) | Use the YOLO v3 algorithm to locate the head area | (1) | In a complex construction scene, it can be judged whether the hardhat exists in the screen and whether it is worn on the head area | (1) | The function of the algorithm is still not powerful enough and needs to be extended (such as the recognition function of personnel, etc.) |
| | | | (2) | Calculate the color pixels of general helmets | | | | |
| | | | (3) | Assignment | | | | |
| | | | (4) | Calculate the confidence of wearing standard | (2) | Real-time performance is very good | | |
| | | | (5) | Compare the test results | | | | |

These studies show that CV-based object detection technology can effectively monitor the PPE use of workers at construction sites. The technology also provides early warning in time and obtains the photos and videos of construction sites through multi-directional cameras. This will not affect the construction process, and the scope of its monitoring is very wide. It becomes convenient to not have managers walking around and patrolling. In addition, from the statistical machine learning method that relies on feature detector + SVM to the deep learning method that depends on CNN, its accuracy and speed of target detection are also improving; nevertheless, it still has some technical limitations and challenges, such as insufficient in-depth understanding of the scene, some visual masking problems and the inaccurate recognition and detection of small targets (e.g., protective gloves, goggles, etc.).

### 4.1.2. Posture Recognition during Construction

In previous studies, many scholars stated that the inappropriate working postures of construction workers are the main cause of safety accidents [19,38]. Behavior-based safety (BBS) has become a trend in safety research. Traditional BBS requires managers to conduct human observation and on-site monitoring to understand unsafe behavior and postures that cause accidents [2,10,11]. Nevertheless, this manual observation method has some limitations, such as high costs and low efficiency [10]. The computer vision behavior monitoring methods can help to address these limitations. For this kind of research, CV-based action recognition technology has achieved remarkable results [10,50].

Workers are a dynamic subject at construction sites, and they perform different activities and have varied action patterns (e.g., bending, lifting, climbing). It is of great importance to identify these actions for the purpose of effective safety management. To prevent false detection of human bodies appearing in the static background area, Peddi [51] proposed a human action recognition method based on the background subtraction. Although this method is not restricted by certain conditions (e.g., light source), the image quality ob-

tained is rough (Figure 10). Combined with the follow-up research of Seo et al. [38] and Liu et al. [52], this behavior detection method can be divided into four steps, including tracking the main body of workers, using the algorithm model to check the background perform segmentation, using histograms to extract features and using classifiers to classify data.

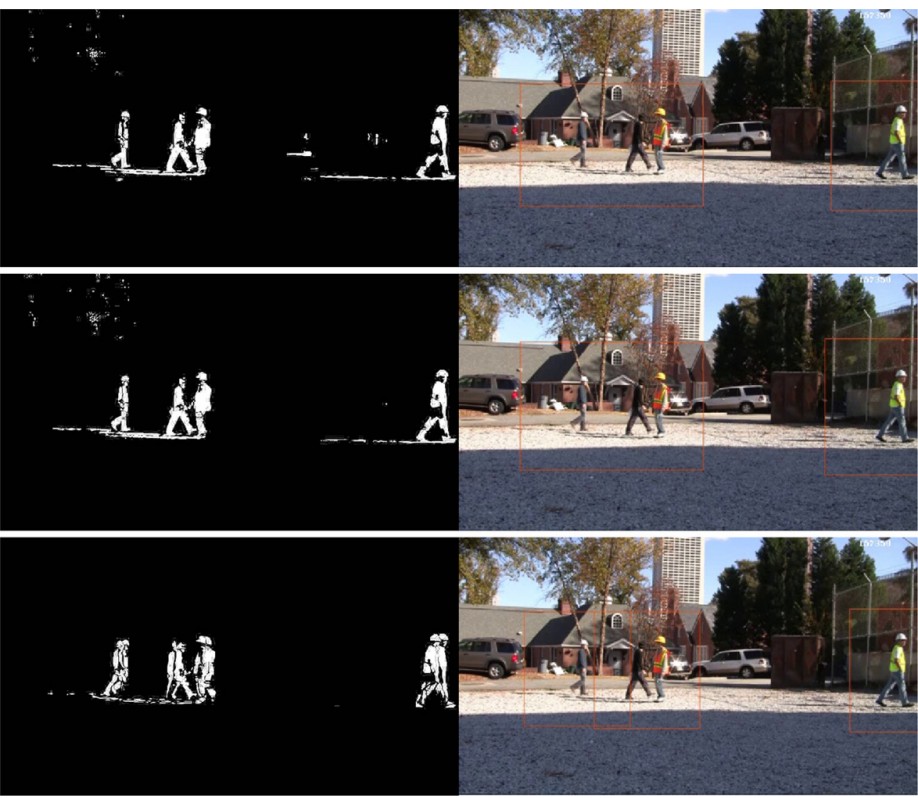

**Figure 10.** Background subtraction legend. Reproduced with permission from ref. [45]. Copyright 2012 Elsevier.

While CV-based deep learning has not been widely used, researchers used depth images and stereo cameras to obtain dynamic image information of workers so as to obtain higher-resolution images. In particular, the use of Kinect and RGB-D motion sensors has enabled researchers to extract clear and rich human motion information. Different from two-dimensional images, researchers can capture more details about the postures of different parts through the three-dimensional images. The most representative one is the extraction of the 3D human skeleton model proposed by SangUK Han [53]. Han et al. [53] proposed a basic framework for motion classification, which contains three basic elements, including three-dimensional motion information data collection, feature extraction and motion classification. This framework is the foundation of the subsequent research on the motion classification prediction. Many subsequent studies used the method of extracting 3D human skeleton model from motion data to further analyze and process the data and classify, identify and predict the workers' actions [11,35,50,54]. The process can be divided into five steps, including extracting 3D motion data information (Figure 11), reducing the dimensionality of the motion data (dimensionality reduction), using a suitable model such as Gaussian Process Dynamic Model (GPDM) to model the average trajectory of samples in low-dimensional space, using related algorithms (e.g., dynamic time warping) to measure the distance between the average trajectory and the motion data set, and classifying actions based on distance (support vector machine SVM is generally used).

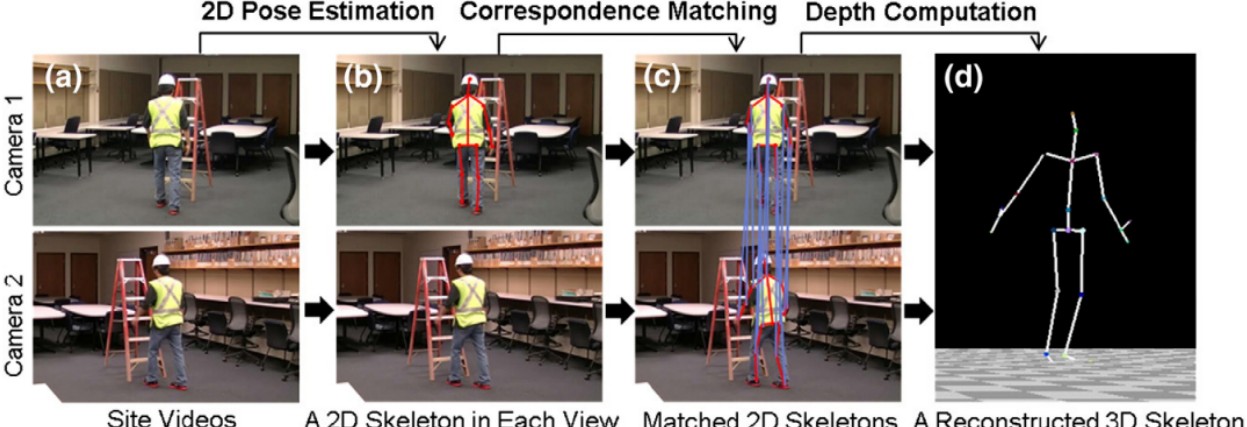

**Figure 11.** Skeleton capture process of 3D motion feature. (**a**) Two videos from a 3D camera or two separate cameras, (**b**) Estimate the position of body joints on 2D image sequences and 3D reconstruction. (**c**) Converting 2D body joints to 3D coordinates, (**d**) Getting a 3D skeleton model. Reproduced with permission from ref. [11]. Copyright 2013 Elsevier.

Nowadays, deep learning has been used to explore the behavior recognition of construction workers [30,54–56]. In the field of deep learning, the development of various neural networks has made the recognition of workers' actions more automated. A close examination of the historical studies found that some common deep learning methods, such as a convolutional neural network (CNN), a deep neural network (DNN) and a recurrent neural network (such as LSTM), have been applied in the field of worker behavior recognition. For instance, Zhang et al. [56] and Chu et al. [57] used 2D camera to obtain images and combined them with multi-stage CNN to extract 3D joint information so as to make classification judgment on workers' postures. Ding et al. [30] proposed a hybrid deep learning model based on CNN and LSTM to automatically identify workers' unsafe behavior. Son et al. [58] proposed the use of depth residual network (Resnet-152), which is one of the classic CNN models, to detect construction workers accurately and quickly in different poses and background in image sequences. Kong et al. [59], Yu et al. [60], Yu et al. [61] and Yu et al. [62] proposed an automatic workload evaluation method by combining CV-based deep learning with intelligent insole pressure sensor and biomechanical analysis. Zhao et al. [63] proposed the use of a DNN model to identify the postures of construction workers based on the motioning data captured by the wearable inertial measurement unit (IMU) sensor. Table 2 summarizes the research in the field of worker behavior recognition.

**Table 2.** Research details of behavior recognition.

| Reference | Algorithm Model | Methods | | Type of Data | Contributions | | Limitations | |
|---|---|---|---|---|---|---|---|---|
| Seo et al. [38]; Liu et al. [52] | Statistical Machine Learning | (1)<br>(2)<br><br><br>(3)<br><br><br>(4) | Online tracking<br>Background subtraction and Region of Interest (ROI)<br>Feature extraction based on shape and radial histogram<br>Sports classification (using K-Nearest Neighbor or SVM) | 2D image | (1)<br><br><br><br><br><br>(2) | The classification accuracy of unsafe postures is better than human observation<br>The practical performance is also good | (1)<br>(2) | The image is not clear enough<br>The ability to distinguish between different postures still needs to be improved |

**Table 2.** *Cont.*

| Reference | Algorithm Model | Methods | Type of Data | Contributions | Limitations |
|---|---|---|---|---|---|
| Han and Lee [11]; Seo et al. [35]; Han et al. [50]; Han et al. [64] | Statistical Machine Learning | (1) Extract 3D human skeleton model from motion data; common motion capture systems include VICON, JVC 3D Everio Camcorder, Microsoft Kinect Senor, RGB-D Senor (2) Kernel PCA is usually used to reduce the dimensionality of motion data (3) Model the average trajectory of samples in low-dimensional space, such as using Gaussian Process Dynamics Model (GPDM) (4) Use the DTW algorithm to measure the distance between the average trajectory and the motion data set (5) Use a classifier to classify according to distance (SVM is the main) | 3D image | (1) The visual capture system is easy to use and low cost (2) Uninterrupted labor movement (3) Wide tracking range (4) The detection accuracy of unsafe actions is high, especially when combined with joint direction information data, the accuracy is as high as 99.5% | (1) Sensitive to light source, not suitable for outdoor construction detection (2) The accuracy of the 3D skeleton extracted from the video needs to be verified (3) Various types of unsafe behaviors need to be tested (4) Two-dimensional pose estimation needs to verify the generalized training data set (5) Privacy issues of video recording |
| Zhang et al. [56];Chu et al. [57] | Deep Learning | (1) Use a single 2D camera to obtain a 2D skeleton (2) Using multi-stage CNN structure to extract 3D joint positions and joint angles as classification features (3) Train the postures of the arms, back and legs and perform classification evaluation | 3D image | (1) The recognition accuracy of the three body parts is as high as 98.6%, 99.5% and 99.8% (2) This method can realize reliable and accurate efficacy evaluation | (1) Errors in the position information of some joints and bones will cause classification errors |
| Ding et al. [30] | Deep Learning | (1) Use CNN to extract visual features from video (2) Sort the learning features supported by the LSTM model | 2D image | (1) Ability to automatically extract and classify unsafe behaviors (2) The accuracy of behavior detection exceeds the current state-of-the-art method | (1) Further understanding of the background of spatio-temporal information is needed (2) Need to pay attention to the actions of multiple equipment/workers in the video frame at the same time |

**Table 2.** *Cont.*

| Reference | Algorithm Model | Methods | Type of Data | Contributions | Limitations |
|---|---|---|---|---|---|
| Son et al. [58] | Deep Learning | (1) Extract feature maps through the deep residual network (ResNet-152) <br> (2) Bounding box regression and labeling of the original image through Faster regions with CNN feature (R-CNN) | 2D image | (1) Using ResNet can accurately and quickly detect multiple workers in the image without relying on limited assumptions about the worker's posture, appearance and background | (1) Accuracy, precision and recall rate still need to be improved |
| Kong et al. [59]; Yu et al. [60]; Yu et al. [61]; Yu et al. [62] | Deep Learning | (1) Use DL algorithm (hourglass network) to estimate three-dimensional joint coordinates <br> (2) Estimate external load based on plantar pressure data <br> (3) Estimation of joint bearing capacity based on anthropological parameters <br> (4) Calculate joint torque based on joint three-dimensional coordinates and external load <br> (5) Evaluate workload based on joint torque and joint capacity | 3D image | (1) Combining CV, pressure sensor technology and biomechanical analysis, a new automatic workload assessment method is proposed | (1) There is still a certain error in the measurement of the joint position |
| Zhao et al. [63] | Deep Learning | (1) Using a DNN model that integrates CNN and two LSTM layers, it can automatically perform feature engineering and sequential pattern detection | 2D image | (1) The convolutional LSTM model is better than the traditional ML-based model | (1) Insufficient sample size <br> (2) Model performance still needs to be improved |

According to these studies, CV-based action recognition has developed rapidly in recent years. From the background-subtraction-based rough estimation to the development of the depth camera and the current depth learning methods, the capture of human postures is becoming more and more accurate. At the same time, with the addition of time information, real-time detection has also been greatly improved. However, the motion postures of human body are changeable, and the current motion data set cannot include all of these postures. In addition, the measurement of motion vectors involving human bones and joints will produce certain errors (e.g., rotation angle, spatial orientation), which will affect the detection accuracy. The research in this field still faces many challenges.

## 4.2. Interaction between Workers and External Environment

A construction site is a dynamic and complex system, which is characterized by the interaction of construction workers with the external environment that includes construction equipment, materials and other objects [19]. When workers interact with the external environment in an inappropriate manner, they expose themselves to dangerous environments [10]. Historical investigations revealed that around 58% of occupational safety accidents are caused by construction equipment collisions [65] and about 40% of them are caused by falls from heights [66,67]. These two types of accidents are the most common ones at construction sites. It is a research hotspot to explore the use of computer technologies in effectively monitoring and pre-controlling these two types of accidents at construction sites.

### 4.2.1. Monitoring of Collision Accidents

Hinze et al. [68] found that collision accidents are associated with equipment, workers and environment, and the authors stated that the combined effects of these three had a significant impact on the occurrence of collision accidents. Zhang et al. [69] pointed out that the two main factors that lead to collision accidents include close contact between workers and construction equipment and the overcrowding of workers and equipment during construction. Researchers use real-time positioning and tracking of workers and construction equipment to detect their locations in order to measure their proximity. When there is a potential inappropriate spatio-temporal relationship, there will be real-time warnings provided to workers to minimize the occurrence of collision accidents. In this process, resource location and tracking technology has become the core. To prevent construction accidents, previous studies have also explored the use of sensor technologies (e.g., GPS, RFID, UWB) to determine the proximity between workers and equipment and compare preset thresholds to detect the risk of collision [69]. The construction site has a large area and includes a large number of people, and the installation of sensors is time-consuming and costly. Because of the low cost and applicability of CV-based object tracking technology, it has been widely used in the monitoring of such accidents.

The monitoring of collision accidents is usually to detect and track the entities (e.g., workers, equipment) at construction sites and determine the potential danger caused by proximity or crowding. When CV-based deep learning has not been widely used, researchers tend to combine video cameras with HOG + color feature description, HOF optical flow histogram, SIFT and other methods to detect the existence of building site entities. This method heavily relies on the manual feature extraction from traditional machine learning and pattern recognition [17]. With the development of deep learning (especially CNN) technology, the monitoring of entities in building scenes has gradually become automated [70].

Based on the CV and fuzzy reasoning, Kim et al. [71,72] proposed a safety assessment system in the moving entity collision accident scene. The system uses image acquisition and wearable devices to detect and track a scene entity and evaluates the safety level of each object based on fuzzy reasoning, which provides early warnings to workers through the danger information displayed by the visualization module. Based on the research findings of Kim et al. [71], Zhang et al. [69] fused CV-based deep learning with the fuzzy reasoning process. Kim et al. [73,74] proposed a visual monitoring method based on unmanned aerial vehicle (UAV) to automatically measure the proximity between construction units, which can detect the risks around workers in advance through UAV + computer vision to facilitate timely intervention. Tang et al. [75] and Cai et al. [76] designed a context-aware LSTM method that used visual data with rich context information to predict workers' trajectories. This model integrated individual movement information and context information (including entity movement information, work group information and potential destination information). Jeelani et al. [77,78] combine eye-tracking technology with CV, collect the workers' gaze points on three-dimensional point clouds by using wearable eye movement instruments, automatically locate their gaze points to analyze their viewing

behavior and calculate their attention distribution. This method of using workers' first perspective (FPV) is helpful to design safety measures and strengthen safety training. Jeelani et al. [79] applied the deep learning algorithm to the semantic segmentation of the visual scene around workers, which improves the accuracy of danger detection. Yan et al. [37] proposed a three-dimensional space congestion estimation method, which generates a 3D space from 2D video frames for proximity and congestion calculations. Son et al. [80] proposed a real-time early warning system that used monocular cameras on both sides of heavy equipment to acquire data in three dimensions (3D) and estimate the location of workers to detect possible collisions. Fang et al. [81] combined semantics and prior knowledge into monocular vision to derive the location information of construction-related entities at construction sites. By using the excavator as an example, Yuan et al. [17] used the three-dimensional tracking and positioning technology to prevent workers from moving close to hazards. Guo et al. [82] detected the dense vehicles in UAV images by using the CNN end-to-end method. Luo et al. [83] proposed the use of CV and deep learning technologies to track the location and operation status of different types of building equipment in surveillance video and designed an automatic estimation framework. Table 3 summarizes the research on the collision risk between workers and construction entities.

**Table 3.** Research details of collision between workers and construction entities.

| Reference | Test Purpose | Methods | | Contributions | | Limitations | |
|---|---|---|---|---|---|---|---|
| Kim et al. [71,72] | On-site safety assessment for collision accidents of moving entities | (1) (2) (3) | Use GMM as background subtraction Kalman filter for target tracking Fuzzy theory set to simulate the reasoning process of experts | (1) (2) (3) | Fusion of CV technology and fuzzy reasoning Automatic utilization of professional safety knowledge The interaction of multiple risk factors can be displayed through the visualization module | (1) (2) | The vision processing algorithm still needs to be improved, and the detection accuracy and tracking consistency need to be improved Affected by light and occlusion |
| Zhang et al. [69] | On-site safety assessment for collision accidents of moving entities | (1) (2) | The Faster-R-CNN model constructs fast regions for detection Use the Matlab fuzzy inference toolbox to take the proximity and congestion in the digital image as the main information | (1) (2) (3) (4) | Fusion of CV technology and fuzzy reasoning Set thresholds to improve collision risk management capabilities Non-contact measurement Automatically identify workers and equipment through images | (1) (2) (3) | The effect of real-time warning is not good The evaluation is based on a 2D scene, which is deviated from the real 3D scene Equipment types are not diversified enough |
| Kim et al. [73] | Proximity analysis through UAV system | (1) (2) | Deep neural network YOLO-v3 for target positioning Develop an image correction method that allows to measure the actual distance of the 2D image collected from the drone | (1) (2) | Estimated average absolute distance error <0.9 m, average absolute percentage error 4% Able to predict the collision hazard around workers in advance | (1) (2) | The calculation efficiency of the rectification method needs to be improved Need to build an IoT cloud integration system |

**Table 3.** *Cont.*

| Reference | Test Purpose | Methods | | Contributions | | Limitations | |
|---|---|---|---|---|---|---|---|
| Tang et al. [75] | Predict the path of workers and equipment to prevent proximity collision hazards | (1) | Long-term prediction through the embedding of LSTM and contextual clues | (1) | The displacement error is smaller than the model that only depends on the target movement | (1) | Not suitable for long-distance trajectory prediction |
| | | (2) | Collect large-scale trajectory data set Voyager to verify | (2) | The pixel value of the positioning error is also very small | (2) | The accuracy of the location still needs to be improved |
| Jeelani et al. [77,78] | Wearable technology real-time tracking and danger warning | (1) | Use SIFT feature extraction and 3D reconstruction to pre-identify the region of interest | (1) (2) | Save time High degree of automation and high recognition accuracy | (1) | The experiment time is too short |
| | | (2) | Analyze gaze behavior and identify predefined hazards | | | | |
| Jeelani et al. [79] | Wearable technology real-time tracking and danger warning | (1) (2) | Worker positioning Use Mask-R-CNN to semantically segment the visual scene around workers | (1) | The detection accuracy rate of workers approaching danger exceeds 93% | | The system cannot determine the distance between the worker and the dynamic dangerAny visible danger in the frame will trigger an alarm |
| Fang et al. [81] | Proximity analysis of related entities | (1) | Use Mask-R-CNN to identify and segment related entity structures | (1) | The positioning errors for excavators, workers and steel piles are reduced to 0.367, 0.132 and 0.148 m respectively | (1) | The occlusion of key parts will affect the results |
| | | (2) | Using a priori knowledge model to estimate the location of equipment, workers and materials in various scenarios | | | | |
| Yan et al. [37] | Analysis of workers' congestion in construction scenes | (1) | Faster-R-CNN detects workers | (1) | Average error of two-dimensional proximity between workers <0.4 m | (1) | The problem of occlusion |
| | | (2) | Worker 3D joint position estimation | | | (2) | Long-distance measurement problem |
| | | (3) | Extraction of view invariant features to estimate the distance between the worker and the camera lens | | | | |
| Yuan et al. [17] | Heavy equipment positioning to prevent approach hazards | (1) | Use optical flow estimation for edge extraction and two-dimensional detection | (1) | Provide a template generation method based on geometric shapes and motion constraints to detect architectural entities | (1) | Different tracking speed and distance will affect drift error |
| | | (2) | 2D tracking and 3D triangulation | | | (2) | The problem of occlusion of nodes |
| Guo et al. [82] | Congestion of equipment (dense vehicles) | (1) | DL-based end-to-end network OAFF-SSD for feature detection | (1) | Adding the fusion feature module, the detection accuracy is higher | (1) | Real-time performance is not good enough |

These studies prove that the CV-based resource tracking and positioning technology can obtain the time track and real-time positions of detection objects at construction site. It helps to avoid the constraints of traditional resource trackers and plays important roles in the analysis of workers' proximity risk and crowding. However, a construction site is a dynamic system, which is more complex than expected. This means that there are difficulties in the measurement of trajectory and positions, and different kinds of errors may occur. In addition, when the attributes of tracking objects are complex (e.g., tracking a group, a specific position of the body), it performs not well. These problems should be addressed in future studies.

### 4.2.2. Monitoring Fall Accidents

The main cause of fatal injuries is the fall of workers from high places. A close examination of historical studies found that there are few explorations about monitoring the falling accidents by using the CV technology. Kolar et al. [67] stated that as unprotected edges are the cause of workers falling from height, the focus should be on real-time detection of safety barriers. This study proposed a CNN-based security guard rail detection model. The data set is generated and trained by adding background images to the three-dimensional model of guardrail. The basic feature extraction of neural network is constructed by using VGG-16 and verified by images. Fang et al. [84] pointed out that the main reason for falling and falling from height are that workers walk on unstable engineering structures (e.g., steel bars, concrete). To solve this, Fang et al. [84] developed an automatic CV method, and this method uses a Mask-R-CNN to detect individuals who pass through the structure support during construction. The method includes the two modules: (1) the Mask-R-CNN module used to detect structural support and personnel and an (2) overlapping detection module used to identify the relationship between human and structural support.

## 5. Research Challenges and Future Study

Although CV technology has been widely used to monitor the unsafe behavior of workers at construction sites and has made a considerable contribution to the improvement of construction site safety, its development and application in this field still face some challenges. This sector discusses the common problems in the use of CV technology to monitor the unsafe behavior of construction workers. On this basis, this study suggests potential solutions and future research directions.

### *5.1. Object Detection Level*
#### 5.1.1. Deeper Scene Understanding

Through sorting and summarizing previous studies, it was found that object detection technology can detect some specific and interested project entities from images of construction sites and further evaluate their possible unsafe behaviors. However the understanding of scenes is often not comprehensive in project management. The object detection technology can only detect objects in certain scenes, and the understanding of "the whole scene" is still complex and challenging. This requires the in-depth mining, extraction, understanding and reasoning of scene semantic information, and few studies explored such areas.

To obtain more comprehensive scene understanding information, scholars should further combine CV technology with some professional knowledge theory sets, such as fuzzy reasoning [69]. Scholars should also reveal the interaction between multiple risks through more comprehensive visualization module information. In addition, 3D reconstruction technology (e.g., the 3D point cloud technology) can also obtain more comprehensive scene space information [77,79].

#### 5.1.2. Visual Occlusion Problem

Visual occlusion is the most common problem when using CV technology. When a worker is partially or completely obscured by some objects, most vision-based methods cannot detect and monitor the worker. In addition, when the worker's back is towards

the camera, the joints of his body's limbs will be blocked by his own body. This is the "self-occlusion" phenomenon [11]. In this situation, some postures of the worker cannot be accurately identified and classified.

There are two currently proposed solutions to deal with this problem, including adjusting the camera positions and increasing the number of cameras [16]. By adjusting the camera positions (e.g., placing the camera as high as possible), the whole body of a worker can be observed as comprehensively as possible, and this can reduce the chance of occlusion. In addition, by placing multiple cameras at construction sites, a wide site coverage can be provided. This helps to reduce some monitoring blind spots. However, when a construction site is crowded with workers and equipment, the multi-camera method becomes ineffective. The second type of method is adopting more optimized deep learning methods (e.g., Faster-R-CNN) [10]. Even if some workers' movements are not detected immediately due to partial occlusion, the relatively fast processing speed of such algorithms can still detect these workers' movements in the next video frame [15]. However, it should be noted that even the best algorithms still cannot detect some occluded entities accurately due to the constraints of technologies, which should be one of the future research directions in this field.

### 5.1.3. Detection of Small Objects

The current object detection technology is able to capture some objects in construction scenes, and its detection accuracy is also very high. However, when it detects some objects with small unit volume, it shows low accuracy. Researchers changed the distance between cameras and workers to solve this problem. Thus, cameras are placed near workers to detect small objects [50,64]. However, this method cannot solve the fundamental problem and can only monitor a small range of workers.

Most researchers believed that improving the resolution of video images is the key to solve this problem. When images are clearer, it will be easier for researchers to capture some small targets. Therefore, future research should focus on reconstructing the corresponding high-resolution images from the observed low-resolution images. This means there is a need of more in-depth research on the "super-resolution technology" [85–87].

### *5.2. Action Recognition Level*
### 5.2.1. Larger Action Sample Size

Many historical studies mentioned the problem of sample size and data set. When workers are performing construction activities, their movement postures are constantly changing, and there are also diverse types of action involved. Nowadays, most of the algorithm models contain relatively fixed target motion in the motion data set [88]. For example, the data set selected by Kim and Cho contains 14 kinds of target motions. Although this motion data set contains various target motions, more than half of them are walking-related [88]. There are many other movements that have been ignored.

Therefore, future research needs to collect more general and larger data sets from actual construction workers. Deep networks can capture sufficient visual features from data sets [89]. Especially by using long-term and short-term memory (LSTM) model [30], deep networks can obtain time series data sets containing action repetition and duration information. Therefore, with the support of DL method, the motion image information contained in the worker motion data sets will be more comprehensive.

### 5.2.2. Problems of Detection Accuracy

In recent years, with the continuous development of deep learning, the accuracy of worker action recognition is getting higher and higher. Nevertheless, the measurement of human bodies will inevitably produce errors, especially the measurement of some motion vectors involving human bone joints, such as joint rotation angle, rotation direction and bearing pressure and load. This will have impacts on detection accuracy.

Therefore, it is suggested that future research can continue to explore more advanced depth learning algorithms and combine them with some other technologies (e.g., biomechanical analysis, pressure sensor, gravity accelerometer, etc.) [59,60,62,90–92] to realize more automatic human physical information extraction and more accurate pose estimation.

### 5.3. Object Tracking Level

### 5.3.1. Construction of 3D Space

When tracking moving entities at construction sites, many researchers prefer viewing construction sites as a 2D plane, as most of the motion modes are linear. Nevertheless, as a construction site is a 3D space, the danger faced by workers may come from all directions. Consequently, there will be corresponding errors in the prediction of distance and trajectory.

Therefore, the future research should be devoted to tracking target entities in the context of 3D space and applying the 3D sensing equipment to the monitoring system [69], which helps to detect the surrounding hazards in a more comprehensive manner.

### 5.3.2. Irregularity of Object Motion

In the process of object tracking, some physical properties of tracked objects affect the tracking accuracy due to the irregularity of motions.

The first is the deformation of tracking objects. In the process of moving objects, its appearance will change constantly. At this time, the filter is constantly updated, and the updated filter cannot guarantee to fully track the target of the next frame, which usually leads to tracking drift [93]. Therefore, it is suggested that future research focus on two aspects: (1) constantly updating the apparent model of objects to adapt to the changes of the apparent model and (2) control the update of filters.

The second is the scale transformation of tracking objects. Scale transformation refers to the phenomenon of scale change from far to near or from near to far during the movement of targets. Predicting the size of a target frame is also a challenge in object tracking. How to predict the scale change coefficient of the target quickly, accurately, and directly affects the accuracy of tracking [94]. In response to such problems, there are some methods. For instance, when generating candidate samples in the motion model, a large number of candidate boxes with different scales can be generated. Or, objects on multiple targets with different scales can be tracked, which helps to generate multiple prediction results and select the best one as the final prediction target. These methods also point out directions for further research in the future.

In addition, there is also the problem of motion blur, which refers to the blurring of a target area caused by the movement of objects or cameras. This will make the tracking effect poor. To deal with such a problem, the mean shift tracking method can perform well [95]. It can obtain information from fuzzy motions and complete the object tracking task. Therefore, to address this problem, the future research can still focus on the CV + mean shift algorithm. It is worth mentioning that some new tracking algorithms need to be developed.

### 5.4. Some Supplements at Other Levels

Construction activities are complex and dangerous, which depends on the coordination of various parties. Many countries educate their construction workers safety skills through training programs and check safety knowledge through qualification and certification tests. However, there are still many construction workers who have not received the corresponding certification, which poses various threats to construction safety [96,97]. In future research, it is suggested to use vision-based personal identification to address such challenge. Face recognition is carried out through the visual system. Once workers who have not obtained the corresponding safety certification or construction workers who violate safety regulations are found, the system will provide real-time warnings and formulate relevant punishment measures. This will help to strengthen construction safety.

Therefore, in future research, CV-based worker identification and construction qualification certification technology should also be developed that can not only prevent irrelevant personnel from entering construction sites but also ensure a high level of construction safety.

## 6. Conclusions

Computer vision (CV) technology has been used to monitor the unsafe behavior of construction workers. Using bibliometrics and content-based literature analysis methods, this paper provides a comprehensive literature review of historical related studies in this field. On this basis, the study points out the existing limitations and challenges and suggests future research directions.

First of all, this study briefly described background on developing CV technology and the roles of CV technology in monitoring construction sites. In addition, the collected studies were statistically analyzed by using the bibliometric methods, and the research trends and hotspots in this field were also clarified. Moreover, the historical research contents were explained from two perspectives, including workers' own perspective and the perspective of the interaction between workers and construction environment. In summary, the study reveals the existing research on the application of CV technology in monitoring the unsafe behavior of construction workers in a systematic and comprehensive manner.

Although the application of CV-based object detection, object tracking and action recognition technologies have made progress in the field of construction safety monitoring, it still has limitations and challenges. In terms of object detection, research on scene understanding, visual occlusion and small target detection is still not deep enough. Regarding object tracking, its accuracy is affected by the complex 3D building environment and the irregularity of tracking object motions. In terms of action recognition, the lack of sample size of motion data sets and the accuracy of human posture capture often bring confusion to relevant researchers. In view of these limitations and challenges, the corresponding research directions have been suggested. It is hoped that these limitations and challenges can be solved in future research.

It is expected that this article will not only enhance stakeholders' understanding the use of CV technology in monitoring the unsafe behavior of construction workers but also provide valuable insights for the CV-based safety and health management in practice.

**Author Contributions:** Conceptualization, Q.M. and W.L.; methodology, Q.M. and W.L.; software, W.L.; validation, Q.M., W.L. and X.H.; formal analysis, Q.M. and W.L.; investigation, Q.M. and W.L.; resources, W.L.; data curation, W.L.; writing—original draft preparation, Q.M. and W.L.; writing—review and editing, Q.M., W.L., Z.L. and X.H.; visualization, Q.M. and W.L.; supervision, Q.M.; project administration, Q.M. and Z.L.; funding acquisition, Q.M. and Z.L. All authors have read and agreed to the published version of the manuscript.

**Funding:** This work was supported by the National Natural Science Foundation of China (Nos. 72071096, 71971100, 71671078); Social Science Fund of Jiangsu Province (19GLB005, 19GLB018); The Key Project of Philosophy and Social Science Research in Colleges and Universities in Jiangsu Province (2018SJZDI052); sponsored by Qing Lan Project of Jiangsu Province; Key Research Base of Universities in Jiangsu Province for Philosophy and Social Science "Research Center for Green Development and Environmental Governance".

**Institutional Review Board Statement:** Not applicable.

**Informed Consent Statement:** Not applicable.

**Data Availability Statement:** The data presented in this study are available on request from the corresponding author.

**Conflicts of Interest:** The authors declare no conflict of interest.

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
