# Peer review of "Applications of Computer Vision in Monitoring the Unsafe Behavior of Construction Workers: Current Status and Challenges"

_buildings, doi:10.3390/buildings11090409_

Round 1
Reviewer 1 Report
The topis of the research is up to date, the paper itself appers to have a scientific potential.
Suggesstion to athe author/authors to be considered:
Introduction part:
- Introduction part may be extended with overview of world statistics related to accidents at construction sites/industry, not only the data from China. It will emphasize te importance of the topic.
- Definition of the term "unsafe bahvior" should be provided - classification of such behaviour in construction sites, possible with relation to the accidents and their consequences.
Background part:
- Application examples of CV in practice and industry could be represented in this part briefly. Benefits, negatives, the cost of implementation, technical requrements of using the CV on construction sites.
- The issue of creating and using of risky scenarios for CV - interaction with scene, construction processes and techniques, objects, other workers, machines should be introduced briefy.
- Also, the issue of measuring and inerpreting tthe probability and impact of risky/unsafe behaviour could be discussed.
Research methods part:
- Aim of the research should be formulated and stated exactly in this part.
- Method presented should be related to the expected research results and aim of the study - reorganize this part more systematicly to get clear formal overview on your research methods.
- Clarify, why the Wos database of publications was used as the source of information.
Content based literature review part:
- Make sure the subsections in this chapter are linked to the methodology stated in previous part, and backgrounf part.
- Summary part of results and discussion part should be added - it will be helpfull for the reader after presenting this part literature review. And it tends to be a standart of research papers. At least the summarization of results and limitations of sample studies for its practical implications
Challenges and future directions part:
- Limitations of your study shold be added.
- Arguments for future research suggestions shoud be represented - based on which facts and results your suggestion for future directions are presented.
Conclusion part:
- The contribution and most important findings of your study should be identified in this part.
Author Response
Response to Reviewer 1 Comments
Dear reviewer, thank you for your comments on this article. We feel very honored. According to your suggestion, we have improved the article. The following is our response, please review it:
Introduction part:
Comment 1:
Introduction part may be extended with overview of world statistics related to accidents at construction sites/industry, not only the data from China. It will emphasize the importance of the topic.
Response 1:
Dear reviewer, according to your suggestion, we added the census data of fatal and nonfatal building injury accidents counted by the U.S. Bureau of Labor Statistics in 2015 and 2018. The statistics of these two documents are very detailed, which can clearly show the number and proportion of construction accidents. The following are links to these two documents:
http://www.doc88.com/p-9743834946518.html
https://www.docin.com/p-1733522010.html?docfrom=rrela
At the same time, we have also modified the contents of the original text. The following are the modified contents (corresponding to lines 26-32 in the article):
Occupational safety in the construction industry is a global problem, not unique to any country. According to the census data of the U.S. Bureau of labor statistics, there were 970 and 965 fatal construction injury accidents in the United States in 2016 and 2017, accounting for about 19% of all occupational deaths in that year. In addition, the incidence of nonfatal occupational injuries and diseases in the construction industry is also 30% higher than the industry average, especially some fall injuries and musculoskeletal diseases.
Comment 2:
Definition of the term "unsafe behavior" should be provided - classification of such behavior in construction sites, possible with relation to the accidents and their consequences.
Response 2:
Dear reviewer, by consulting relevant materials in this field, we found the definitions of "unsafe behavior of workers" and "unsafe state" in two documents, which are respectively:
- Petersen, D.C. HUMAN ERROR REDUCTION AND SAFETY MANAGEMENT. University of Northern Colorado., 1980.
- Meng, Q.F.; Liu, W.Y.; Li, Z.; Hu, X. Influencing Factors, Mechanism and Prevention of Construction Workers' Unsafe Behaviors: A Systematic Literature Review. Int. J. Env. Res. Public Health 2021, 18, 22, doi:10.3390/ijerph18052644.
At the same time, we explained the definition in the text and made some modifications to this part, corresponding to lines 38 to 42. The modifications are as follows:
The unsafe behavior of construction workers refers to the dangerous behavior of violating organizational discipline, operating procedures and methods in their professional activities, and the unsafe state refers to the material conditions that lead to accidents, including material and potential hazards in the working environment. These hazards are often caused by human operation, that is, the unsafe behavior of workers.
Background part:
Comment 1:
Application examples of CV in practice and industry could be represented in this part briefly. Benefits, negatives, the cost of implementation, technical requirements of using the CV on construction sites.
Response 1:
Dear reviewer, according to your suggestion, we revised the original text. First, we briefly described the industry application of CV technology, its advantages, costs, etc. we added it at the beginning of chapter 2.2 as a basic description and guidance. Lines 173-182 in the corresponding text are as follows:
At present, the research on CV technology in the construction industry mainly focuses on building structure monitoring and productivity analysis, and there is less research on unsafe behavior identification. The traditional identification and control of unsafe behaviors mainly rely on manual methods, and the effect is poor. In particular, a large number of images taken by the monitoring camera cannot be processed automatically and effectively. The development of CV technology just provides effective support for the automatic identification of unsafe behaviors. In particular, CV technology does not need to attach equipment to workers. It not only greatly reduces costs, but also has little impact on workers. At the same time, it can also process a large number of image data in time and quickly, so CV technology is suitable for the construction site environment.
Next, we briefly described the shortcomings of CV technology and the technical requirements for its application to the construction site. We added it to the end of chapter 2.2 to connect the following. The following contents are the modified contents, corresponding to lines 190-196 in the text.
However, CV technology itself is limited to extracting unsafe information and cannot further evaluate information to identify unsafe behaviors and conditions. Therefore, the unsafe behavior monitoring method based on CV technology should not only consider the extraction of construction information, but also combine with existing policies and relevant experience. It requires a more systematic framework to discuss how CV technology is applied to complex construction sites.
Comment 2:
The issue of creating and using of risky scenarios for CV - interaction with scene, construction processes and techniques, objects, other workers, machines should be introduced briefly.
Response 2:
Dear reviewer, based on your suggestions, we described the scene-based method in more detail in the background section. The following content is our improvement, corresponding to lines 202-210 of the article.
First, the scene-based approach is to understand and evaluate any potential risks in a static scene by examining the scene in a safe context. Scene understanding refers to the integration of the information of the various components of the construction site. Its main purpose is to understand "what is in the scene (e.g., people, materials, machines, etc.)". Therefore, the object detection technology has been applied here. This technology searches the image through the known object model, so that the object of interest can be detected based on the semantic information. Only if the project entity of interest is confirmed, can the follow-up more in-depth research be carried out. In general, the scene-based approach is the first step and the cornerstone of the entire research.
Comment 3:
Also, the issue of measuring and interpreting the probability and impact of risky/unsafe behavior could be discussed.
Response 3:
Dear reviewer, according to your suggestions, we have improved and systematically discussed this problem. We briefly explain why CV based methods can increase the probability of discovering unsafe behaviors and their impact. The following is what we added, corresponding to lines 224-246 in the text.
To sum up, at the construction site, CV based methods are divided into three categories: object detection, object tracking and action recognition, which makes it possible to intelligently monitor unsafe behaviors and conditions at the construction site.
Object detection can be directly used to identify some unsafe behaviors and conditions of construction sites. The most common method is to divide a captured large image window into small spatial areas for analysis, extract features from small areas, and then classify them. From manual extraction to automatic extraction, from SVM to CNN, its speed and accuracy are constantly improving. The probability of discovering unsafe behavior is also greatly increased.
Object tracking can create the time track of the detected object when moving in the scene and master its real-time position. There are two main kinds of current research, namely CV based 2D tracking and 3D tracking. 2D tracking mainly tracks the target by matching the feature points and shape contours in the video frame, while 3D tracking mainly uses 3D tracking sensors to establish 3D coordinates to obtain movement information. From the perspective of space, this method can comprehensively detect some unsafe approaching behaviors of workers in space.
Action recognition is the process of labeling action labels on the image. This method can extract human features from the image, such as shape and time motion, which is conceptually similar to the feature extraction of target detection. But it is more complicated because some specific motion vectors are added (e.g., joint position, joint angle, etc.). This method can better extract some small actions. These three methods can monitor the construction site well, find the unsafe behaviors of workers in multiple directions, and make a great contribution to the construction safety management.
Research methods part:
Comment 1:
Aim of the research should be formulated and stated exactly in this part.
Comment 2:
Method presented should be related to the expected research results and aim of the study - reorganize this part more systematically to get clear formal overview on your research methods.
Responses to 1 and 2:
Dear reviewer, thank you for your suggestions. According to your suggestion, we have made changes. This part is mainly a brief overview of the research methods and the preparation process of the required reference materials. So we first revised the title to "research methods and material preparation". Next, we will integrate the first two suggestions you put forward and make improvements. The following are our improvements, corresponding to lines 247-260 in the article.
The main purpose of this paper is to comprehensively reveal the research status of CV technology in the field of monitoring unsafe behavior of construction workers, and carry out a more comprehensive literature review. Therefore, this study decided to adopt the comment method based on content analysis. This method is a recognized method of literature review, synthesis and result rationalization, which has been unanimously recognized by the majority of researchers. In this chapter, through a more systematic bibliometric analysis, the academic relationship and research hotspots of computer vision in the field of building safety are mapped, the research theme is highlighted and determined, and the previous research framework and context are corroborated. At the same time, through the selection of topics and research fields, periodical screening and other processes, the applicability and quality of obtaining literature are ensured, which makes good preparations and paves the way for the specific content-based analysis in the next chapter.
Comment 3:
Clarify, why the WOS database of publications was used as the source of information.
Response 3:
Dear reviewer, according to your request, we explained why we use WOS database as a reference source. The following is our reply, corresponding to lines 262-271 in the article.
Web of Science has a powerful analysis function, which can quickly lock high impact papers and find the research directions concerned by global peer authorities, especially the Science Citation Index Expanded (SCIE) and Social Science Citation Index (SSCI) in the core collection of WOS. These two academic journal paper citation index databases contain the most comprehensive high impact academic journals in the world. In addition, the conference proceeding Citation Index- Science (CPCI-S) in the core collection of WOS covers the annual meeting minutes of various industry authorities, which is also very leading edge and guiding. Based on the above reasons, SCIE, SSCI and CPCI-S databases in the core collection of WOS are finally determined as reference sources.
Content based literature review part:
Comment 1:
Make sure the subsections in this chapter are linked to the methodology stated in previous part, and background part.
Response 1:
Dear reviewer, according to your suggestion, we have added relevant sentences in each section of this chapter to correspond to the method described in the background section. As follows:
Lines 329-331: Researchers often use image-based object detection technology to monitor the PPE use of construction workers.
Lines 391-392: For this kind of research, the action recognition technology based on CV has achieved remarkable results.
Lines 473-474: Because of the low cost and applicability of object tracking technology based on CV, it has been widely used in the monitoring of such accidents.
Comment 2:
Summary part of results and discussion part should be added - it will be helpful for the reader after presenting this part literature review. And it tends to be a standard of research papers. At least the summarization of results and limitations of sample studies for its practical implications.
Response 2:
Dear reviewer, according to your request, we have made a summary of each part. The specific contents added are as follows:
Object detection part (Lines 382-391):
These studies show that the object detection technology based on CV can effectively monitor the PPE use of workers on the construction site, give early warning in time, and obtain the photos and videos of the construction site through multi-directional cameras, which will not affect the construction process, and the scope of its monitoring is also very wide. It is very convenient without managers walking around and patrolling. In addition, from the statistical machine learning method based on feature detector + SVM to the deep learning method based on CNN, its accuracy and speed of target detection are also improving, but it still has technical limitations and challenges, such as insufficient in-depth understanding of the scene, some visual masking problems, and inaccurate recognition and detection of some small targets (e.g., protective gloves, goggles, etc.).
Action recognition part (Lines 456-464):
Through these studies, it is not difficult to find that CV based action recognition has developed rapidly in recent years. From the rough estimation based on background subtraction to the development of depth camera, and then to the current depth learning methods, its capture of human posture is more and more accurate. At the same time, with the addition of time information, its real-time detection is also greatly improved, However, the motion posture of human body is changeable, and the current motion data set cannot include it all. In addition, the measurement of motion vectors involving human bones and joints will produce certain errors (e.g., rotation angle, spatial orientation, etc.), which will affect the detection accuracy. The research in this direction still faces many challenges.
Object tracking part (Lines 536-544):
These studies prove that the resource tracking and positioning technology based on CV can obtain the time track and real-time position of the detection object in the construction site. It gets rid of the constraints of the traditional resource tracker and plays a great role in the analysis of workers' proximity risk and crowding. However, the construction site is a dynamic whole, which is much more complex than expected, which often has certain difficulty in the measurement of trajectory and position, and will produce large errors. In addition, when the attributes of the tracking object are complex, such as tracking a group, a specific position of the body, its performance is not very good. These problems are worth further thinking and discussion.
Challenges and future directions part:
Comment 1:
Limitations of your study should be added.
Comment 2:
Arguments for future research suggestions should be represented - based on which facts and results your suggestion for future directions are presented.
Responses to 1 and 2:
Dear reviewer, according to your suggestions, we have made improvements and more detailed division and supplement to the whole chapter 5. We changed the title of Chapter 5 to "Research challenges and future study", in which some challenges are some problems and limitations found in the research. The following are our improvements to the whole chapter 5. Corresponding to lines 561-693 in the article.:
- Research challenges and future study
Although the CV technology has been widely used to monitor the unsafe behavior of workers at construction sites and has made a considerable contribution to the improvement of construction site safety, its development and application in this field still have some challenges. This chapter discusses the common problems in the use of CV technology to monitor unsafe behaviors of construction workers. On the basis, the study suggests potential solutions and proper future research directions.
5.1. Object detection level
5.1.1. Deeper scene understanding
Through sorting and summarizing previous studies, it is found that object detection technology can detect some specific and interested project entities from the images of the construction site, and further evaluate their possible unsafe behaviors. However, in terms of project management, the understanding of the scene is often not comprehensive. For the current object detection technology. It can only detect some objects in some scenes, not "the whole scene", and the understanding of "the whole scene" is very complex. This requires in-depth mining, extraction, understanding and reasoning of scene semantic information. At present, there is little research in this field.
In future research, in order to obtain more comprehensive scene understanding in-formation, scholars can further combine CV technology with some professional knowledge theory sets, such as fuzzy reasoning [69], and display the interaction between multiple risks through more comprehensive visualization module information. In addition, 3D reconstruction technology (e.g., 3D point cloud technology) can also obtain more comprehensive scene space information [77,79].
5.1.2. Visual occlusion problem
Visual occlusion is the most common problem while using the CV technology. When a worker is partially or completely obscured by some objects, most vision-based methods cannot detect and monitor the worker. In addition, when the worker’s back is towards the camera, the joints of his body’s limbs will be blocked by his own body. This is what Han et al. call “self-occlusion” [11]. In this situation, some postures of the worker cannot be accurately identified and classified.
The current proposed solutions to this problem are mainly proposed from two perspectives, including adjusting the camera position and increasing the number of cameras [16]. By adjusting the camera position (e.g., placing the camera as high as possible), the whole body dynamics of a worker can be observed as comprehensively as possible and this can reduce the chance of occlusion. In addition, by placing multiple cameras at construction sites, a wide site coverage can be provided. This helps to reduce some monitoring blind spots. However, when a construction site is crowded with workers and equipment, multi camera method becomes ineffective. The second type of method is to adopt more optimized deep learning methods (e.g., Faster-R-CNN) [10]. Even if some workers’ movements are not detected immediately due to partial occlusion, the relatively fast processing speed of such algorithms can still detect these workers’ movements in the next video frame [15]. However, it should be noted that even the best algorithms still cannot detect some occluded entities accurately due to the constraints of technologies, which should be one of the future research directions in this field.
5.1.3. Detection of small objects
The current object detection technology has been able to better capture some objects in the construction scene, and its detection accuracy is also very high. However, when it detects some objects with small unit volume, it shows low accuracy. At first, researchers tried to change the distance between the camera and workers to solve this kind of problem. Cameras are placed near workers to detect small objects [50,54]. However, this method cannot solve the fundamental problem, and can only monitor a small range of workers.
Most researchers believe that improving the resolution of video image is the key to solve this kind of problem. Only when the image is clearer can we better capture some small targets. Therefore, future research should focus on reconstructing the corresponding high-resolution image from the observed low-resolution image, that is, to conduct more in-depth research on the so-called "super-resolution technology"[85-87].
5.2. Action recognition level
5.2.1. Larger action sample size
Many historical studies mentioned the problem of sample size and data set. When workers are performing construction activities, their movement postures are constantly changing and there are also diverse types of action involved. Nowadays, most of the algorithm models contain relatively fixed target motion in the motion data set [88]. For example, the data set selected by Kim and Cho contains 14 kinds of target motion, although this motion data sets contains various target motions, more than half of them are walking-related [88]. There are many other movements that have been ignored.
Therefore, future research needs to collect more general and larger data sets from the actual construction workers. Deep network can capture enough visual features from data sets [89]. Especially, by using long-term and short-term memory (LSTM) model [30], it can obtain time series data sets containing action repetition and duration information. There-fore, with the support of DL method, the motion image information contained in the worker motion data set will be more comprehensive.
5.2.2. Problems of detection accuracy
In recent years, with the continuous development of deep learning, the accuracy of worker action recognition is getting higher and higher, but the measurement of human body will inevitably produce errors, especially the measurement of some motion vectors involving human bone joints, such as joint rotation angle, rotation direction, bearing pressure and load, which will have a certain impact on the detection accuracy.
Therefore, we believe that future research can continue to explore more advanced depth learning algorithm models from this perspective, and combine some other technologies (e.g., biomechanical analysis, pressure sensor, gravity accelerometer, etc.) [60,61,63,90-92], so as to realize more automatic human physical information extraction and more accurate pose estimation.
5.3. Object tracking level
5.3.1. Construction of 3D space
When tracking the moving entities at construction sites, many researchers prefer viewing the construction site as a 2D plane as most of the motion modes are linear. Nevertheless, as a construction site is a 3D space, the danger faced by workers may come from all directions. Consequently, there will be corresponding errors in the prediction of distance and trajectory.
Therefore, the future research should be devoted to tracking target entities in the con-text of 3D space and applying the 3D sensing equipment to the monitoring system [69], which helps to detect the surrounding hazards in a more comprehensive manner.
5.3.2. Irregularity of object motion
In the process of object tracking, some physical properties of the tracked object will affect the tracking accuracy due to the irregularity of motion.
The first is the deformation of the tracking object. In the process of moving the object, its appearance will change constantly. At this time, the filter is constantly updated, and the updated filter cannot guarantee to fully track the target of the next frame, which usually leads to tracking drift. Therefore, future research should focus on two points: (1) Constantly update the apparent model of the object to adapt to the changes of the apparent model; (2) Control the update of the filter.
The second is the scale transformation of the tracking object. Scale transformation refers to the phenomenon of scale change from far to near or from near to far during the movement of the target. Predicting the size of the target frame is also a challenge in object tracking. How to predict the scale change coefficient of the target quickly and accurately directly affects the accuracy of tracking. In response to such problems, the common methods are: when generating candidate samples in the motion model, generate a large number of candidate boxes with different scales, or track objects on multiple targets with different scales, generate multiple prediction results, and select the best one as the final prediction target. At the same time, it also points out the direction for further research in the future.
In addition, there is also the problem of motion blur, which refers to the blurring of the target area caused by the movement of the object or the camera. This will make the tracking effect poor. For such problems, the mean shift tracking method shows good results. It can get information from fuzzy motion and complete the object tracking task. Therefore, for this problem, our future research may still focus on CV + mean shift algorithm. It is worth mentioning that some new tracking algorithms need to be developed.
5.4. Some supplements at other levels
Construction activities are complex and dangerous, which depends on the coordination of various trade work. Many countries educate their construction workers safety skills through training programs and check safety knowledge through qualification and certification tests. However, there are still many construction workers who have not received the corresponding certification, which poses a serious threat to the safety of construction sites [93,94]. In the future research, using vision-based personal identification can be an important way of addressing such challenge. Face recognition is carried out through the visual system. Once workers who have not obtained the corresponding safety certification or construction workers who violate the safety regulations are found, the system will pro-vide real-time warnings and formulate relevant punishment measures, which strengthens the safety construction at construction sites.
Therefore, in future research, CV based worker identification and construction qualification certification technology should also be further developed, which can not only prevent irrelevant personnel from entering the construction site, but also ensure the professional level of construction.
Conclusion part:
Comment 1:
The contribution and most important findings of your study should be identified in this part.
Response 1:
Dear reviewer, according to your suggestion, we reorganized our conclusion. The following is our modification, corresponding to lines 695-725 in the article:
The computer vision (CV) technology has been used to monitor the unsafe behavior of construction workers. Using bibliometrics and content-based literature analysis methods, this paper makes a detailed literature review of the historical literature in this field, points out the existing limitations and challenges, and puts forward the future research direction.
Before the systematic review, this article briefly describes the background environment of the development of CV technology and the role of CV technology in the process of monitoring the construction site, so as to pave the way for the following analysis to a certain extent. Then, the collected literature is statistically analyzed by bibliometric methods, and the research trends and hotspots in this field are clarified. Then with workers as the main body, the research content is explained from two perspectives from the inside to the outside, namely: (1) the worker's own perspective; (2) the perspective of the interaction between the worker and the construction environment. This paper comprehensively reveals the relevant research status of CV in the monitoring of unsafe behavior of workers on the construction site, and comprehensively combs and summarizes the research of existing CV technology in the supervision of unsafe behavior of workers on the construction site based on multiple dimensions such as algorithm model, method, data type, advantages and disadvantages.
Although CV based object detection, object tracking and action recognition technology has achieved great results in the field of building safety monitoring, it still has limitations and challenges and faces great challenges. In terms of object detection, the research on scene understanding, visual occlusion and small target detection is not deep enough. In terms of object tracking, its accuracy is affected by the complex 3D building environment and the irregularity of tracking object motion. In terms of action recognition, the lack of sample size of motion data sets and the accuracy of human posture capture often bring confusion to relevant researchers. In view of these limitations and challenges, we put forward the corresponding research directions in turn. It is hoped that these limitations and challenges can be solved in future research.
It is expected that this article will not only enhance stakeholders’ understanding about using CV in monitoring unsafe behaviors of construction workers but also provide valuable insights for the computer vision-based safety and health management in practice.
Finally, we thank you again for your comments on our article. Your suggestions will help us better improve the level of our article.

Reviewer 2 Report
I have reviewed the manuscript, which provides quite comprehensive review of the application of computer vision in construction health and safety.
The manuscript is well organized. Providing a review on the subject guides a reader towards the aim of the paper.
The discussion of the research findings has been robustly presented.
Few issues to address to make this paper suitable for publication:
The authors are recommended to professionally proofread the manuscript to improve its readability and avoid language issues.
Clearer justification is required for the reason behind the use of only WoS as an appropriate database for searching the related papers.
Research and practical implications as well as the research limitations need to be covered in the concluding remarks of the paper, which is currently missing.
Author Response
Response to Reviewer 2 Comments
Dear reviewer, thank you for your comments on this article. We feel very honored. According to your suggestion, we have improved the article. The following is our response, please review it:
Comment 1:
The authors are recommended to professionally proofread the manuscript to improve its readability and avoid language issues.
Response 1:
Dear reviewer, according to your suggestions, we have further proofread and improved the translation of the paper, and we have uploaded the improved manuscript to the system.
Comment 2:
Clearer justification is required for the reason behind the use of only WOS as an appropriate database for searching the related papers.
Response 2:
Dear reviewer, according to your request, we explained why we use WOS database as a reference source. The following is our reply, corresponding to lines 262-271 in the article.
Web of Science has a powerful analysis function, which can quickly lock high impact papers and find the research directions concerned by global peer authorities, especially the Science Citation Index Expanded (SCIE) and Social Science Citation Index (SSCI) in the core collection of WOS. These two academic journal paper citation index databases contain the most comprehensive high impact academic journals in the world. In addition, the conference proceeding Citation Index- Science (CPCI-S) in the core collection of WOS covers the annual meeting minutes of various industry authorities, which is also very leading edge and guiding. Based on the above reasons, SCIE, SSCI and CPCI-S databases in the core collection of WOS are finally determined as reference sources.
Comment 3:
Research and practical implications as well as the research limitations need to be covered in the concluding remarks of the paper, which is currently missing.
Response 3:
Dear reviewer, according to your suggestion, we reorganized our conclusion. The following is our modification, corresponding to lines 695-725 in the article:
The computer vision (CV) technology has been used to monitor the unsafe behavior of construction workers. Using bibliometrics and content-based literature analysis methods, this paper makes a detailed literature review of the historical literature in this field, points out the existing limitations and challenges, and puts forward the future research direction.
Before the systematic review, this article briefly describes the background environment of the development of CV technology and the role of CV technology in the process of monitoring the construction site, so as to pave the way for the following analysis to a certain extent. Then, the collected literature is statistically analyzed by bibliometric methods, and the research trends and hotspots in this field are clarified. Then with workers as the main body, the research content is explained from two perspectives from the inside to the outside, namely: (1) the worker's own perspective; (2) the perspective of the interaction between the worker and the construction environment. This paper comprehensively reveals the relevant research status of CV in the monitoring of unsafe behavior of workers on the construction site, and comprehensively combs and summarizes the research of existing CV technology in the supervision of unsafe behavior of workers on the construction site based on multiple dimensions such as algorithm model, method, data type, advantages and disadvantages.
Although CV based object detection, object tracking and action recognition technology has achieved great results in the field of building safety monitoring, it still has limitations and challenges and faces great challenges. In terms of object detection, the research on scene understanding, visual occlusion and small target detection is not deep enough. In terms of object tracking, its accuracy is affected by the complex 3D building environment and the irregularity of tracking object motion. In terms of action recognition, the lack of sample size of motion data sets and the accuracy of human posture capture often bring confusion to relevant researchers. In view of these limitations and challenges, we put forward the corresponding research directions in turn. It is hoped that these limitations and challenges can be solved in future research.
It is expected that this article will not only enhance stakeholders’ understanding about using CV in monitoring unsafe behaviors of construction workers but also provide valuable insights for the computer vision-based safety and health management in practice.
Finally, we thank you again for your comments on our article. Your suggestions will help us better improve the level of our article.

Reviewer 3 Report
I would like to express great appreciation to the authors of this research. This research was very well planned, carried out and implemented. The structure of the paper is correct. The obtained research results are very interesting and valuable for practical use. I agreee that the computer vision (CV) technology could be used to monitor the unsafe behavior of construction workers.
Only one suggestion: - according to the Instructions for Authors https://www.mdpi.com/journal/sustainability/instructions): References style must be changed - according to the template.
Author Response
Response to Reviewer 3 Comments
Dear reviewer, thank you for your comments on this article. We feel very honored.
According to your suggestion, we downloaded the reference template at the website you gave, imported it into endnote, and updated the reference format in the article. We have uploaded the improved article and hope you can review it further.
Finally, thank you again for your valuable suggestions, which makes our article format more standardized.

Round 2
Reviewer 2 Report
The authors have addressed all my comments.